# CAPP-130: A Corpus of Chinese Application Privacy Policy Summarization and Interpretation

**Pengyun Zhu**[1], **Long Wen**[1], **Jinfei Liu**[1,2]*, **Feng Xue**[1], **Jian Lou**[1,2], **Zhibo Wang**[1], **Kui Ren**[1]

[1] Zhejiang University
[2] ZJU-Hangzhou Global Scientific and Technological Innovation Center
{22121092, wenlongcs, jinfeiliu✉, henryxue, jian.lou,
zhibowang, kuiren}@zju.edu.cn

## Abstract

A privacy policy serves as an online internet protocol crafted by service providers, which details how service providers collect, process, store, manage, and use personal information when users engage with applications. However, these privacy policies are often filled with technobabble and legalese, making them "incomprehensible". As a result, users often agree to all terms unknowingly, even some terms may conflict with the law, thereby posing a considerable risk to personal privacy information. One potential solution to alleviate this challenge is to automatically summarize privacy policies using NLP techniques. However, existing techniques primarily focus on extracting key sentences, resulting in comparatively shorter agreements, but failing to address the poor readability caused by the "incomprehensible" of technobabble and legalese. Moreover, research on Chinese application privacy policy summarization is currently almost nonexistent, and there is a lack of a high-quality corpus suitable for addressing readability issues. To tackle these challenges, we introduce a fine-grained CAPP-130 corpus and a TCSI-pp framework. CAPP-130 contains 130 Chinese privacy policies from popular applications that have been carefully annotated and interpreted by legal experts, resulting in $52,489$ annotations and $20,555$ rewritten sentences. TCSI-pp first extracts sentences related to the topic specified by users and then uses a generative model to rewrite the sentences into comprehensible summarization. Built upon TSCI-pp, we construct a summarization tool TSCI-pp-zh by selecting RoBERTa from six classification models for sentence extraction and selecting mT5 from five generative models for sentence rewriting. Experimental results show that TCSI-pp-zh outperforms GPT-4 and other baselines in Chinese application privacy policy summarization, demonstrating exceptional readability and reliability. Our data, annotation guidelines, benchmark models, and source code are publicly available at *https://github.com/EnlightenedAI/CAPP-130*.

## 1 Introduction

A privacy policy is a legal document that outlines how the service providers collect, process, store, manage, and use the personal information of individuals who interact with its applications. This pivotal agreement is designed to safeguard personal data from misuse, thwart breaches of personal privacy, and uphold data security. Service providers are exclusively permitted to collect, process, store, manage, and use personal information in accordance with the stipulations outlined in the agreement, and strictly with the explicit consent and authorization given by users.

---

*Corresponding Author.

37th Conference on Neural Information Processing Systems (NeurIPS 2023) Track on Datasets and Benchmarks.

The study of privacy policies began in the 1960s when Alan Westin first defined privacy as control over personal information in his book [43] and proposed the need for its protection. With the emergence of the internet and the development of social media and e-commerce, the importance and complexity of privacy policies have been constantly increasing, becoming an important area of legal and technological research. Today, Privacy policies serve as a crucial agreement between users and service providers. However, most of the privacy policies suffer from poor readability and are often deemed "incomprehensible" [13, 35]. This is largely due to their inclusion of complex technobabble and legalese, as well as their penchant for intricate grammar structures, such as multiple negatives. Moreover, there is a severe trend of privacy policies becoming increasingly lengthy and "incomprehensible" [3]. This leads to a substantial number of users who tend to bypass reading privacy policies by clicking on "Agree" or "Join Now" [50]. Regrettably, this practice may inadvertently expose users to potential risks related to their privacy information, as these privacy policies may contain terms that conflict with existing laws [26]. This clearly goes against the original intent of the privacy policy design and can even be seen as a facade allowing service providers to excessively collect and use personal information.

To address the challenge of making privacy policies more comprehensible, several initiatives like LPL [14], TILT [15], PPL [41], and P3P [9] have emerged towards establishing a standardized and formalized guideline for privacy policies. These endeavors aim to aid service providers in presenting their privacy policies in a clear and comprehensible manner, thus enabling users to comprehend these privacy policies more effectively. Nonetheless, ensuring the genuine compliance of service providers to these guidelines remains an outstanding challenge.

Another approach to address the issue of incomprehensible privacy policies is through summarization, as explored in studies by [18, 17]. This technique involves using concise formats, clear titles, and understandable language to condense content [50]. However, service providers often are unwilling to invest cost and effort in providing privacy policy summarization. An effective solution is to use NLP technology to automatically summarize important content from privacy policies. However, previous studies [24, 18, 33, 39] focus on extracting sentences related to certain topics from privacy policies, but overlook the "incomprehensible" issues caused by the use of uncommon technobabble or legalese. Furthermore, to the best of our knowledge, existing privacy policy corpora, such as the English corpus OPP-115 [44] and the Chinese corpus CA4P-483 [49], only annotate data practices and their corresponding categories. These annotations are useful for identifying paragraphs containing specific data practices. However, no existing corpus provides comprehensible interpretations for such paragraphs to address the challenge of "incomprehensible". In conclusion, there is a current lack of both corpora and corresponding tools to tackle the issue of privacy policies being "incomprehensible".

In this paper, we address the challenge by presenting two main contributions in the areas of privacy policy corpora and summarization tools. First, we provide a Chinese Application Privacy Policy Corpus (CAPP-130) that has been carefully annotated and revised by legal experts. It contains 130 Chinese application privacy policies with $38,889$ sentences, $52,489$ annotations, and $20,555$ rewritten sentences. Specifically, based on Chinese law provisions and the characteristics of Chinese application privacy policies, our legal expert team develops a set of annotation guidelines and personally annotates the CAPP-130 corpus according to these guidelines. In these guidelines, the legal experts categorize data practices into 11 categories and add two special tags for marking sentences that contain data practices or potential privacy risks. They also introduce the rules and examples for sentence rewriting, making it easier to revise sentences with special tags into clear and understandable language. When identifying categories of data practices in privacy policies, the legal experts draw on the experience of OPP-115, using a more refined sentence granularity and allowing a sentence to have multiple tags. Second, we provide a Topic-Controlled Framework for Summarization and Interpretation of Privacy Policy (TCSI-pp). Unlike previous methods that only extract specific sentences, TCSI-pp first retrieves relevant sentences based on the topics chosen from data practice categories by users using a classification model. Then, a generative model is used to rewrite these sentences in a clear and concise manner for the understanding of the general public, with potentially risky sentences emphasized. Finally, we provide a series of benchmarks based on pre-trained models according to TCSI-pp and CAPP-130, including six classification models and five generation models. Among these models, we select the most effective RoBERTa and mT5 to implement the Chinese application privacy policy summary tool (TCSI-pp-zh). Experiments on real privacy policies show that TCSI-pp-zh outperforms GPT-4 and other models, demonstrating higher

readability and reliability in the task of summarizing Chinese application privacy policies. The key contributions of this paper are summarized as follows.

- We create a fine-grained Chinese application privacy policy (CAPP-130) corpus that has been carefully annotated and interpreted by legal experts. CAPP-130 contains $38,889$ sentences across $52,489$ annotations and $20,555$ rewritten sentences.

- We propose a Topic-Controlled Framework for Summarization and Interpretation of Privacy Policy (TCSI-pp).

- We implement a Chinese application privacy policy summarization tool (TCSI-pp-zh) based on TCSI-pp and CAPP-130.

- Experimental results on real chinese privacy policies show that TCSI-pp-zh outperforms GPT-4 and other baselines in Chinese application privacy policy summarization in terms of readability and reliability.

## 2   Related Work

Currently, research on automatic summarization and the construction of corpora for privacy policies primarily concentrates on information extraction, which includes searching for specific *data practices* [7, 18] or answering simple multiple-choice questions [24, 38].

The literature [39] used a rule-based unsupervised method to extract content from five topics in privacy policies, integrating this content according to the themes to form a summary. However, rule-based methods lack flexibility and are unable to draw from past experiences to adapt to new situations. OPP-115 [44] and APP-350 [51] are among the earliest known privacy policy corpora, annotated using expert and crowdsourcing methods respectively. These corpora break down privacy policies into finer data practices, providing useful resources for tasks such as classification and content extraction of privacy policy texts. The research [44, 16, 20] proposed methods to improve extraction or classification in OPP-115. PI-Extract [8] selects 30 privacy policies from OPP-115 and annotates a more detailed text span for Entity Recognition tasks. PrivacyQA [30] and PolicyQA [1] are privacy policy Question Answering System corpus based on OPP-130. These two corpora consist of pairs made up of the "original sentences" of the privacy policy and "questions" of users, which the Question Answering System can extract relevant sentences or paragraphs. Optoutchoice-2020 [6] and Optoutchoice-2017 [34] are two corpora specifically annotated with content related to privacy settings such as ad tracking, which are greatly beneficial for users in extracting and managing privacy settings in privacy policies. Tools [24, 25] defined certain privacy factors and mapped each factor to a risk level using data mining models. Since these tools cannot directly understand the source of the risk, literature [18] re-evaluated the privacy terms at risk and proposed two methods, namely the coverage method and the risk-centered method, for selecting the abstract. Research on Chinese privacy policy is relatively scarce. Recently, we saw the emergence of the first Chinese privacy policy corpus, CA4P-483 [49]. This corpus was annotated by trained university students and provides some classification benchmarks. Moreover, general large language models such as GPT-4 [27], Llama2[40], and ERNIE Bot [32] have demonstrated impressive results in text summarization tasks. However, their performance in specific information retrieval tasks is not as good as fine-tuned models [28], and they are prone to "hallucination" [5, 2]. Studies such as LPL [14], TILT [15], PPL [41], and P3P [9], are committed to establishing a standardized and formalized guideline for privacy policies to mitigate the issue of privacy policies being hard to comprehend. However, the significant challenge of ensuring genuine compliance by service providers with these guidelines remains a huge issue.

In summary, the emergence of these corpora and tools undoubtedly brings more possibilities for research and practice in the field of privacy policies. Although these methods have to some extent reduced the reading difficulty for users, they have not attempted to use new language to summarize and explain the essence of privacy policies, but have directly quoted them verbatim. Such an approach fails to effectively address the problem of incomprehensibility caused by complex sentence structures and professional knowledge. On another note, there is no corpus that has been specifically designed to interpret privacy policies or address the issue of incomprehensibility.

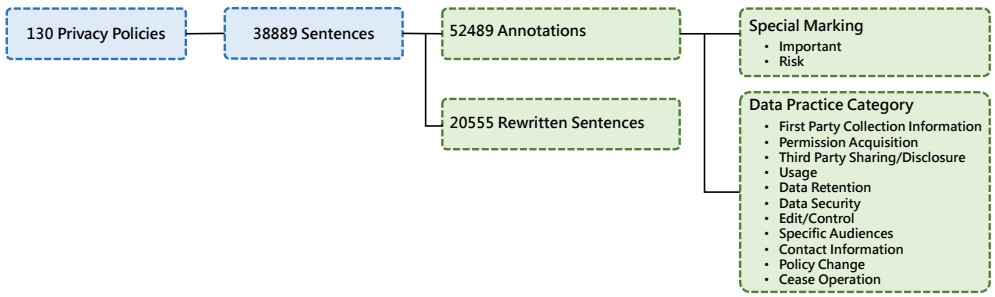

Figure 1: The structure of CAPP-130.

# 3 The CAPP-130 Corpus and TCSI-pp Framework

In this section, we outline the annotation rules, procedures, and in-depth details of CAPP-130 in Section 3.1 and present the proposed summarization framework TCSI-pp in Section 3.2.

## 3.1 The CAPP-130 Corpus

Privacy policies are often obscure and difficult to understand, and they usually require a strong understanding of the subject matter. To ensure that the annotation scheme accurately reflects the actual content of the privacy policy, legal experts have developed a set of annotation guidelines for privacy policy summaries, based on existing work [44]. These guidelines incorporate the characteristics of Chinese privacy policies and legal regulations. The process involved multiple refinements and iterations to reach the final version. Moreover, although automated summarization techniques provide a potential solution to the issue of privacy policies being "incomprehensible", long text automatic summarization is more susceptible to information loss and clause misinterpretation compared to short text. Given this, we utilize finely-grained sentences as samples.

Firstly, the Privacy Policy Summarization annotations guidelines include the complete content required by laws and regulations to map the policy coverage. It consists of three parts: *Data Practice Categories*, *Importance Markings*, and *Rewritten Sentences*. Figure 1 illustrates the structure of CAPP-130.

***Data Practice Category*** also known as *Topic*, is used to describe the category of the sentence or term in privacy policies. Specifically, we have eleven data practice categories as follows.

- **First Party Collection Information**: how and why a service provider collects user information; and impact on service functionality if users refuse to provide information.
- **Permission Acquisition**: how and why a service provider obtains application permissions; and impact on service functionality if users refuse to grant permissions.
- **Third Party Sharing/Disclosure**: how to share or transfer user information with the third party, publicly disclose it, or collect it by the third party.
- **Usage**: how user data is used, including: building data analysis models, personalized content recommendations or service models, automated decision-making models, etc.
- **Data Retention**: how long and where user information is stored.
- **Data Security**: how user information is protected.
- **Edit/Control**: edit and control options available to users(e.g., modify and delete user information, turn off personalized ads or content recommendations, deactivate accounts, etc.
- **Specific Audiences**: practices that pertain only to a specific group of users (e.g., children, Europeans, or California residents)
- **Contact Information**: how contact service provider.
- **Policy Change**: if and how users will be informed about changes to the privacy policy, and inform them what changes have been made.
- **Cease Operation**: how user data will be handled when operation ceased.

Table 1: The statistics of CAPP-130.

| Data Practice Category & Special Marking | Quantity | Percentage | Median | Mean |
|---|---|---|---|---|
| First Party Collection Information | 6967 | 17.9 | 58 | 70 |
| Permission Acquisition | 1852 | 4.8 | 54 | 62 |
| Third Party Sharing/Disclosure | 4740 | 12.2 | 52 | 63 |
| Usage | 3589 | 9.2 | 64 | 75 |
| Data Retention | 1360 | 3.5 | 41 | 46 |
| Data Security | 3000 | 7.7 | 53 | 60 |
| Specific Audiences | 1416 | 3.6 | 54 | 60 |
| Edit/Control | 5324 | 13.7 | 43 | 49 |
| Contact Information | 712 | 1.8 | 41 | 54 |
| Policy Change | 1049 | 2.7 | 35 | 43 |
| Cease Operation | 110 | 0.3 | 64 | 68 |
| Important | 20555 | 52.8 | 52 | 61 |
| Risk | 1815 | 4.7 | 40 | 46 |

***Special Marking*** are used to indicate whether data practices, sensitive personal information, or potential risks are involved. It includes two markings:

- **Important**: Whether it includes one or more of the above data practices.

- **Risk**: Whether it is related to potential data risks.

***Rewritten Sentences*** provide concise and user-friendly interpretations for sentences with "Important'.

Secondly, to ensure the representativeness and diversity of privacy policies, we separately obtained the privacy policies of the top 100 downloaded applications from two major Chinese app markets - *Xiaomi* App Store and *Huawei* App Store. After excluding the privacy policies of duplicate applications and different apps with the same privacy policy, we obtained a total of 130 representative privacy policies. The update date of these privacy policies is up to December 2022. These documents cover various types of mainstream applications (e.g., shopping, live streaming, sports, and navigation) and games (e.g., real-time battles, sports competitions, simulation management, and board games). We used regular expressions to divide the privacy policy into sentence levels.

Finally, we allow each segmented sentence to have multiple labels. Three legal experts annotated the collected privacy policies according to the privacy policy labeling standards, resulting in a high-quality corpus called CAPP-130 (Chinese Application Privacy Policies, set of 130). The corpus comprises $52,489$ annotations for $38,889$ sentences and $20,555$ rewritten sentences. In the annotation process, we randomly select 20 privacy policies, which are jointly annotated by three legal experts, resulting in a Cohen Kappa coefficient [4] of 0.907. This result strongly validates the annotation reliability of each legal expert. On the basis of high reliability, the remaining privacy policies are annotated by one of the legal experts considering the cost performance. For the 20 selected privacy policies annotated by three legal experts, we employ the majority vote mentioned.

The corpus allows us to investigate the composition of typical privacy policies from a data practice perspective. Table 1 presents the statistical information of CAPP-130. *Percentage* refers to the proportion of the number of sentences belonging to the data practice category or special marking to the number of total sentences. As a sentence can have multiple labels, the sum of the percentages may exceed 100%. *Mean* and *median* indicate the average and middle value of sentences lengths. Visually, all categories have mean values above the median, reflecting a right-skewed distribution. The data practice categories *First Party Collection Information* (17.9%), *Edit/Control* (13.7%), and *Third Party Sharing/Disclosure* (12.2%) constitute approximately half of the privacy policy, making them the main components of summaries and areas that require particular attention. Apart from the 52.8% that pertains to content requiring attention in the summary, the majority consists of superficial or introductory statements, contact information, or general information unrelated to user data. Additionally, the rewritten sentences for the *Important* and *Risks* statements have average lengths of 40 and 36, respectively. These figures are lower compared to the average lengths of the original sentences, which were 61 and 46. This indicates an average reduction of approximately 34.4% and 21.7%, respectively.

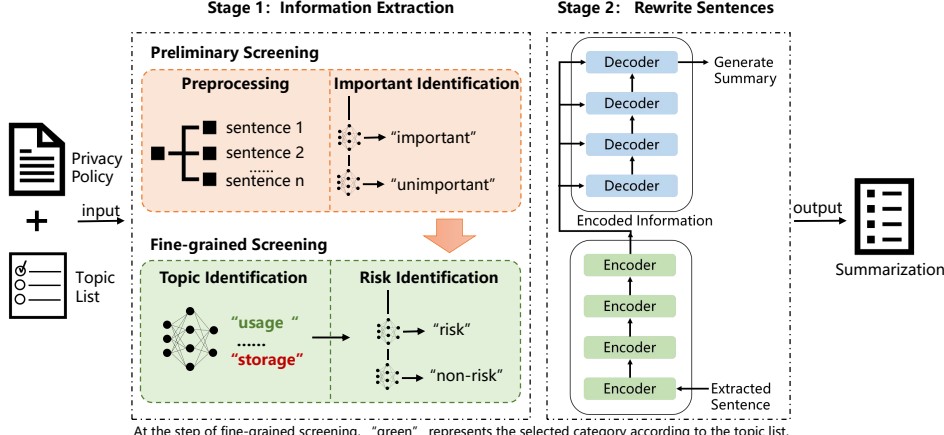

Figure 2: The framework of TCSI-pp.

## 3.2 The TCSI-pp Framework

The *Topic-Controlled Method for Privacy Policy Summarization and Interpretation* (TCSI-pp) framework adopts a hybrid text summarization approach. Algorithm 1 shows the pseudocode of TCSI-pp. Figure 2 presents the basic process flow of this framework. It is mainly composed of Input, Information Extraction, Rewritten Sentences, and Output, the details are as follows:

**Input.** Provide privacy policy text and topics that need summaries.

**Information Extraction.** The information extraction module is applied to extract key information from the privacy policy, consisting of two stages: preliminary screening and fine-grained screening. In the preliminary screening stage, The privacy policy is preprocessed using *regular expressions* to obtain a collection of sentences at the granularity of *sentences*, denoted as $P = \{p_1, \ldots, p_n\}$, where $p_n$ represents the $n$-th sentence of the policy. we input $\forall p$ into the model $F_e(p, \theta_e)$ used for identifying "Important" sentences, where $\theta_e$ is a set of parameters obtained through trained. Fine-grained screening is a multi-class classification model $F_t(p, \theta_t)$ to obtain the topic (data practice category) of $Important$ sentences. we use the For $\forall p | important = true$, $l_i = argmax(F_t(p, \theta_t))$, where $l_i \in L$ corresponds to data practice categories. Based on user-specified multitopics ($MTs$), we extract sentences consistent with them, resulting in the set $Filtered$. A summarization of the Chinese privacy policy typically needs to include the following topics: 'First Party Collection Information', 'Third Party Sharing/Disclosure', 'Usage', 'Edit/Control', and 'Contact Information'. Moreover, we mark $\forall p \in Filtered$ through the "risk" identification model $F_r(s_i, \theta_r)$, highlighting sentences that contain potential risks.

**Rewritten Sentences** Considering the issues of generating false facts and repetitive content in long text generation with auto-regressive models, we also use a transformer-based generative model to rewrite and interpret sentences at the level of data practices. For $\forall p \in Filtered$, we employ the auto-regressive model $P(z_t | p; z_{1:t-1})$ to rewrite the original privacy policy into formulations that are easily comprehensible for the general public.

**Output.** We examine the rewritten sentences based on preselected topics, and highlight sentences with "risk", to enhance their readability.

## 4 Experiments

In this section, we provide the baseline methods of summarization in Section 4.1, examine the performance of the information extraction model in Section 4.2, followed by an exploration of the rewritten sentences model and its performance in Section 4.3, showcase the summarization results of the TCSI-pp-zh tool and existing models in Section 4.4, present a social survey to verify the readability of TCSI-pp-zh in Section 4.5.

**Algorithm 1** TCSI_pp framework.

---

**Input:** Privacy policy $P = \{p_1, ..., p_n\}$; Specified multi topics $MTs \in T$.
**Output:** Summarization $P_{ats}$.
**Initialize:** $Filtered = list()$
#Step 1: Filtering sentences that need to be interpreted.
**For** $\forall p_i \in P$ :
    **If** $F_e(p_i, \theta_e)$ is $True$:     $topics = F_t(p_i, \theta_t)$
    **If** $\exists topic \in topics$ in $MTs$:     $Filtered \leftarrow p_i$
#Step 2: Rewrite into a more understandable sentence.
**For** $\forall ti \in MTs$: $P_{ats} \leftarrow title\_ti$     #Write the title of topic $ti$ to $P_{ats}$
    **For** $\forall s_i \in Filtered$:
        **If** $F_r(s_i, \theta_r)$ is $True$:     $P_{ats} \leftarrow F_{generate}(s_i, \theta_g)$ with highlight
        **Else**:     $P_{ats} \leftarrow F_{generate}(s_i, \theta_g))$
**Return** $P_{ats}$

---

Table 2: Evaluation metrics for the preliminary and fine-grained screening models.

| Method | topic-multi | | important-binary | | risk-binary | |
|---|---|---|---|---|---|---|
| | Micro-F1 | Macro-F1 | Micro-F1 | Macro-F1 | Micro-F1 | Macro-F1 |
| RoBERTa | **0.819** | **0.841** | **0.897** | **0.899** | 0.920 | 0.711 |
| Bert | 0.802 | 0.820 | 0.895 | 0.896 | 0.921 | 0.719 |
| mBERT | 0.809 | 0.821 | 0.889 | 0.889 | 0.918 | 0.709 |
| SBERT | 0.781 | 0.794 | 0.875 | 0.874 | 0.917 | 0.689 |
| PERT | 0.801 | 0.812 | 0.895 | 0.897 | **0.922** | **0.716** |
| ERNIE | 0.807 | 0.821 | 0.895 | 0.896 | 0.921 | 0.702 |

## 4.1 Baseline Methods

For information extraction, we utilize the following models based on the transformer encoder architecture: RoBERTa(*chinese-RoBERTa-wwm-ext*) [10], Bert(*Bert-base-Chinese*) [12], ERNIE(*ernie-3.0-base-zh*) [36], PERT(*chinese-pert-base*) [11], mBERTa(*mbart-large-50*) [37], and SBERT(*SBERT-base-chinese-nli*) [31]. The preliminary screening stage is configured as binary classification, while the fine-grained screening is set up as multi-class classification.

For rewritten sentences, we fine-tune the following models based on the transformer encoder-decoder [42] architecture: mT5 [45], Bert2Bert, Bert2gpt, RoBERTa2gpt, and ERNIE2gpt. These models are initialized with parameters from publicly available models, such as *mT5-small* [45], *Bert-base-Chinese* [12], *ernie-3.0-base-zh* [36], *chinese-RoBERTa-wwm-ext* [10], and *gpt2-base-chinese* [29]. These models can be found on Hugging Face's[2] model repository. To ensure that the input sequence does not exceed the maximum position allowed by the pre-trained models, we truncate the document by removing the ending part. For the complete code and detailed parameters, please refer to our GitHub repository at *https://github.com/EnlightenedAI/CAPP-130*.

## 4.2 Performance Evaluation of the Information Extraction Model

To develop the TCSI-pp information extraction baseline model, we utilized RoBERTa, BERT, ERNIE, PERT, mBERT, and SBERT as the underlying models. Micro-F1 and Macro-F1 were employed as evaluation metrics. During the preliminary screening phase, we fine-tuned the "important-binary" and "risk-binary" classification models based on the "Special marking". For the fine-grained screening model, we conducted fine-tuning using the "Topics". Table 2 displays the evaluation metrics for both the preliminary screening and fine-grained screening models.

In Table 2, it is evident that RoBERTa significantly outperforms other models in the metrics of the "topic" and "important" screening tasks, and its performance in the "risk" screening is nearly on par with the optimal PERT. Therefore, RoBERTa appears to be more suitable for information extraction of CAPP-130. It is worth emphasizing that all models demonstrate a situation where the Micro-F1 is significantly higher than the Macro-F1 in the screening of risk sentence. This is primarily because

---

[2]https://huggingface.co/

Table 3: Evaluation metrics for each topic in fine-grained screening models.

| Topic | RoBERTa | | | mBERT | | |
|---|---|---|---|---|---|---|
| | Precision | Recall | F1 | Precision | Recall | F1 |
| First Party Collection Information | 0.855 | 0.811 | 0.832 | 0.871 | 0.786 | 0.826 |
| Permission Acquisition | 0.809 | 0.866 | 0.836 | 0.858 | 0.812 | 0.834 |
| Third Party Sharing/ Disclosure | 0.832 | 0.847 | 0.839 | 0.835 | 0.863 | 0.849 |
| Usage | 0.644 | 0.520 | 0.575 | 0.760 | 0.325 | 0.455 |
| Data Retention | 0.848 | 0.822 | 0.835 | 0.807 | 0.871 | 0.838 |
| Data Security | 0.905 | 0.676 | 0.774 | 0.869 | 0.641 | 0.738 |
| Specific Audiences | 0.967 | 0.870 | 0.916 | 0.833 | 0.941 | 0.883 |
| Edit/Control | 0.937 | 0.877 | 0.906 | 0.919 | 0.893 | 0.906 |
| Contact Information | 0.988 | 0.921 | 0.954 | 0.965 | 0.921 | 0.943 |
| Policy Change | 0.916 | 0.890 | 0.903 | 0.878 | 0.890 | 0.884 |
| Cease Operation | 0.824 | 0.933 | 0.875 | 0.824 | 0.933 | 0.875 |

Table 4: Evaluation metrics for the rewritten sentence models.

| Model | ROUGE-1 | ROUGE-2 | ROUGE-L | Bert-score | Bart-score | Carburacy |
|---|---|---|---|---|---|---|
| mT5 | **0.753** | **0.609** | **0.733** | **0.888** | **-4.577** | **0.833** |
| Bert2gpt | 0.751 | 0.574 | 0.720 | 0.872 | -4.964 | 0.764 |
| RoBERTa2gpt | 0.749 | 0.577 | 0.719 | 0.872 | -4.975 | 0.755 |
| Bert2bert | 0.718 | 0.535 | 0.689 | 0.864 | -5.020 | 0.747 |
| ERNIE2gpt | 0.623 | 0.406 | 0.581 | 0.809 | -5.716 | 0.715 |

the percentage of risk-related sentences is exceedingly small, and it tends to be relatively implicit. As a result, it becomes more difficult for the models to extract meaningful semantic information.

In order to delve deeper into the performance disparities of the multi-class fine-grained screening models across data practices, Table 3 showcases the Precision, Recall, and F1 scores for RoBERTa and mBERT within all topics.

As shown in Table 3, both the RoBERTa and mBERT models accomplish an evaluation metric exceeding 0.80 in 11 data practice categories, excluding "Usage". Additionally, their scores surpass 0.90 for numerous topics. This is due to the fact that data usage is a relatively sensitive area, and manufacturers often opt for more specialized vocabulary and intricate expressions when describing these sentences compared to other data practice categories. Based on this finding, we recommend that data regulatory authorities further refine the regulatory requirements related to terms of personal information "usage" in privacy policies. It is important to note that the evaluation metrics of both classification models once again confirm the high quality of CAPP-130.

### 4.3 Performance Evaluation of the rewritten sentence Model

In this section, rewritten sentences from CAPP-130 were utilized to fine-tune the models introduced in Section 4.1, serving as benchmarks for the rewriting task. Table 4 displays the *ROUGE* [19], *Bert-score* [48], *Bart-score* [46], and *Carburacy* [23] evaluation metrics for these models:

Table 4 demonstrates that the mT5-based model significantly outperforms other models in terms of all metrics. The ranking is as follows: mT5 >Bert2gpt >RoBERTa2gpt >Bert2bert >ERNIE2gpt.

### 4.4 Performance of the summarization model

TCSI-pp-zh uses RoBERTa as the information extraction model and mT5 as the sentence rewritten model. To showcase the results, we extracted several paragraphs from privacy policies for validation and demonstration purposes. Figure 3 displays the outcomes of GPT-4 and TCSI-pp-zh in summarizing privacy policies, where text having the same background color represents descriptions of the same part of the privacy policy generated by different algorithms; red text emphasizes incorrect content produced in the summary. The Appendix A.3 contains the originally extracted excerpts as well as summarization from other methods, including the human expert, the large language model

| GPT-4: | TCSI-pp-zh: |
|---|---|
| 一、数据采集相关内容 | 1.我们采集的信息与数据 |
| Ⅰ. Relevant Content of Data Collection: | 1. The Information and Data We Collect |
| **1.** 我们收集和使用的个人信息包括您注册册时提供的姓名、性别、年份、地址、电话号码、电子邮箱地址、身份证明文件号码、收费相关的银行信息以及联络人信息等。 | （1）当您注册成为用户或使用我们服务时,收集您提供的姓名、性别、年龄、地址等相关个人信息。 |
| 1. The personal information we collect and use includes the name, gender, year of birth, address, telephone number, email address, identification document number, bank information related to fees,and contact information provided during registration. | (1) When you register as a user or use our services, we collect relevant personal information you provide such as your name, gender, age, and address. |
| **2.**在您使用本网站的服务期间，我们将继续授权使用这些信息，但在注销账号时我们会停止使用这些信息。 | （2）在您注销账号时,我们将停止使用信息。 |
| 2. During your use of our website services, we will continue to have the authorized use of this information. However, upon account cancellation, we will cease the use of this information. | (2) Upon the cancellation of your account, we will cease the use of your information. |
| 3.此外，我们还会自动收集包括设备、软件信息、IP地址、搜索和浏览信息等日志信息。 | （3）您使用我们服务时,系统可能自动采集技术信息。 |
| 3. Additionally, we also automatically collect log information, including device and software info-rmation, IP addres-ses, search and browsing information, and other relevant data. | (3) When you use our services, our system may automatically collect technical information. |
| **4.**为保护您的隐私，我们会采取必要的措施来保护您的个人信息。 | （4）在您接入服务时,收集您的设备或软件信息。 |
| 4.To protect your privacy, we will take necessary measures to safeguard your personal information. | (4) When you access our services, we collect information about your device or software. |
| | （5）收集您在使用我们服务时搜索和浏览的信息。 |
| | (5) We collect information about your searches and browsing history when you use our services. |

Figure 3: Summarizations of GPT-4 and TCSI-pp.

Table 5: Questionnaire statistics of readability.

| Model | Q1 | Q2 | Q3 | Q4 | Q5 | Q6 | Q7 | Q8 | Q9 | Q10 | Percentage |
|---|---|---|---|---|---|---|---|---|---|---|---|
| TCSI-pp-zh | **37** | **29** | **29** | 17 | **29** | 25 | **38** | 24 | 24 | **30** | 46.23% |
| GPT-4 | 20 | 27 | 4 | **39** | 28 | **31** | 19 | **30** | **35** | 24 | 42.13% |
| Llama2 | 4 | 4 | 5 | 3 | 2 | 5 | 4 | 5 | 5 | 5 | 6.39% |
| Checkpoint | 0 | 1 | 23 | 2 | 2 | 0 | 0 | 2 | 2 | 2 | 5.25% |

ERNIE Bot (文心一言) [32], Tongyi Qianwen (通义千问) [21], the generative model Checkpoint (Randeng-Pegasus-523M-Summary-Chinese) [47], and the extractive model TextRank [22].

As illustrated in Figure 3, GPT-4 has the ability to extract specific details from privacy policies and generate summaries. However, it can also produce false content that is convincingly deceptive, containing information not found in the original text (red font). To optimize its output, GPT-4 requires carefully crafted prompts, a skill that most users typically lack. In contrast, TCSI-pp-zh accurately pinpoints key information without needing additional prompts from the user, and its summaries employ more concise language compared to GPT-4. Moreover, as demonstrated in Appendix A.3, TextRank considers nearly every sentence important based on keyword frequency, resulting in excessive extraction. Due to limitations stemming from the nature and distribution of the initial training corpus, Checkpoint fails to effectively extract crucial details from privacy policies. The combined performance of ERNIE Bot and Tongyi Qianwen falls short of GPT-4; they may also generate non-existent text or make inaccurate judgments due to insufficient understanding. Specifically, ERNIE Bot generated content that did not belong to the original text (red font), while Tongyi Qianwen ignored some key sentences (sentences with yellow and purple backgrounds in the original text). It is important to note that the content produced by language models is random and unpredictable across each generation. The text above represents the best-case scenario achieved through repeated prompt optimization, and some unprofessional prompts might even result in bizarre outputs. In comparison, our approach is capable of summarizing and generalizing the extracted content, offering advantages such as accurate information extraction, strong interpretability, and concise understandability. However, there is still a significant difference between TCSI-PP and human privacy policy experts in terms of induction, such as the ability to summarize across paragraphs.

## 4.5 Readability Questionnaire Survey

To evaluate the readability of TCSI-pp-zh, we conducted a social survey on *WJX*[3]. This survey consisted of 10 single-choice questions. Each question included a paragraph from a privacy policy along with four summaries generated by TCSI-pp-zh, CPT-4, Llama2(LLAMA2-Chinese), and Checkpoint (Randeng-Pegasus-523M-Summary-Chinese). We invited interviewees to select the summarization they found most readable and comprehensible. From the survey, We eventually gathered 61 valid responses as shown in Table 5. The detailed statistics of the interviewees are disclosed in Appendix A.4.

As shown in Table 5, among the choices made by the interviewees, 46% rated the readability of the summaries created by TCSI-pp-zh the highest, followed by GPT-4 at approximately 42%, Llama2-Chinese at around 6%, and Checkpoint at around 5%. TCSI-pp-zh received the most votes in six out of the ten questions, positioning it ahead of GPT-4, which only managed to lead in four

---
[3]https://www.wjx.cn/

questions. On the other hand, Llama2-Chinese and Checkpoint did not receive the most votes at all. In conclusion, TCSI-pp-zh demonstrates superior readability compared to GPT-4 and other models when summarizing Chinese privacy policies.

## 5  Conclusion and Future Work

This paper addresses the challenge of understanding complex Chinese application privacy policies which are often filled with technobabble and legalese. To overcome this issue, we propose a *Topic-Controlled Framework for Summarization and Interpretation of Privacy Policy* (TCSI-pp) framework and a fine-grained *Chinese Application Privacy Policy* (CAPP-130) corpus, which has been carefully annotated and rewritten by legal experts. We fine-tune several transformer models using CAPP-130 as a benchmark for data practice classification, summarization, and sentence rewriting. Among these models, we select the most effective RoBERTa and mT5 to implement the Chinese application privacy policy summary tool (TCSI-pp-zh). We also evaluate the summarization performance of TCSI-pp-zh on real privacy policies. Experimental results show that large language models like GPT-4 and other benchmark models tend to suffer from content extraction errors or generate false content. In contrast, TCSI-pp-zh can accurately extract specific privacy terms and present them in concise and understandable language. We present the limitations and discuss the potential solutions as follows. (1) CAPP-130 and TSCI-pp-zh are currently only applicable to Chinese privacy policies. We are currently creating English corpora and models and have plans to extend our efforts to include privacy policies in various other languages. (2) The data collection of CAPP-130 relies on machines, we are moving towards incorporating more human factors in collecting privacy policies for future versions. (3) Although we have made efforts in data annotation and model selection to deal with the possible "illusion" phenomenon of generative models, we still cannot completely eliminate the possibility of information loss and misunderstanding of terms. Building upon the foundation of CAPP-130, we are exploring the implementation of keyword extraction techniques to provide cues for classification models and establish external legal knowledge bases to provide valuable background knowledge for generative models. Furthermore, we are contemplating allowing users to control the level of summarization or offer the choice to view both the summarized and original privacy policies side by side. In doing so, we hope to alleviate the problem of "information loss and misunderstanding". (4) While users can select specific content from the 11 data practices available in our model, they are unable to access information beyond these 11 data practice categories. In our future work, we intend to explore additional data practice categories to enhance the customization options of our summarization tool. (5) Compared with real privacy policy experts, our model still lacks the ability to summarize across paragraphs effectively. Therefore, the effective summarization of lengthy text-based privacy policies is a future research direction for us. (6) We are actively designing a dashboard that will incorporate visual elements such as icons and symbols, with the intention of aligning the summarization process more closely with users' intuitive perceptions.

## Acknowledgments and Disclosure of Funding

The authors would like to thank the anonymous reviewers for their helpful comments. This work was supported in part by the National Key RD Program of China (2021YFB3101100) and NSFC grants (62102352, 62206207).

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
