# CAPP-130: A Dataset of Chinese Application Privacy Policy Summarization and Interpretation

**Pengyun Zhu**[1], **Long Wen**[1], **Jinfei Liu**[1,2]*, **Feng Xue**[1], **Jian Lou**[1,2], **Zhibo Wang**[1], **Kui Ren**[1]

[1] Zhejiang University

[2] ZJU-Hangzhou Global Scientific and Technological Innovation Center

{22121092, wenlongcs, jinfeiliu✉, henryxue, jian.lou,
zhibowang, kuiren}@zju.edu.cn

## A  Appendix

### A.1  Detailed statistics of TCSI-pp Dataset

We split the classified corpus into training data, validation data, and testing data at a ratio of 70:15:15. The rewritten corpus is split into training data, validation data, and testing data at a ratio of 80:10:10. The details of each corpus are shown in Table 1.

Table 1: The statistics of train-validation-test data.

|  | #all samples | #training samples | #validation samples | #testing samples |
|---|---|---|---|---|
| important-binary | 38889 | 27222 | 5833 | 5834 |
| risk-binary | 20505 | 14338 | 3083 | 3084 |
| topic-multi | 20268 | 14190 | 3043 | 3035 |
| rewrite | 19570 | 15656 | 1957 | 1957 |

### A.2  Train Details

We have completed all experiments on two A100s, and the complete code, data, and parameters can be found from *https://github.com/EnlightenedAI/CAPP-130*.

### A.3  Supplementary Experimental

This section provides the original excerpts for privacy policy and verifies the performance of summarization on the other five methods, mentioned in Section 4.4 of the main paper.

Figure 1 shows the excerpts for the original privacy policy used to verify the performance of summarization methods. Figure 2 displays the summarization from a Human expert. Figure 3 displays the actual summarization from two other large language models, ERNIE Bot (文心一言) [5], Tongyi Qianwen (通义千问) [3]. Figure 4 presents the actual results based on the extractive method TextRank [4] and the generative model Checkpoint (Randeng-Pegasus-523M-Summary-Chinese) [6]. The red font indicates content that should not be extracted or does not match the original text.

### A.4  Basic statistical information of the interviewee

Table 2 presents the basic statistical information of the interviewees in the privacy policy readability survey.

---

*Corresponding Author.

37th Conference on Neural Information Processing Systems (NeurIPS 2023) Track on Datasets and Benchmarks.

**Origin:**

一、我们如何收集和使用您的个人信息。

Ⅰ. How we collect and use your personal information:

个人信息是指以电子或者其他方式记录的能够单独或者与其他信息结合识别特定自然人身份或者反映特定自然人活动情况的各种信息。

Personal information refers to various information recorded electronically or by other means that can identify a specific individual or reflect specific activities of an individual.

1、您提供的信息

您注册成为我们的用户或使用我们的服务时，向我们提供的相关个人信息，包括您的【姓名（名称）、性别、年龄、地址、电话号码、电子邮箱地址、身份证件号码、缴费相关的银行信息以及联系人信息等】。您提供的上述信息，将在您使用本网站服务期间持续授权我们使用。在您注销账号时，我们将停止使用上述信息。

When you register as our user or use our services, you provide us with relevant personal information, including your name, gender, age, address, phone number, email address, identification card number, bank information related to payments, and contact information. The aforementioned information you provide will be continuously authorized for our use during your use of this website's services. We will stop using the aforementioned information when you cancel your account.

2、您在使用服务过程中产生的信息

2. Information generated during your use of the service:

（1）日志信息

(1) Log information:

指您使用我们服务时，系统可能通过cookies或其他方式自动采集的技术信息，包括：

Refers to technical information that the system may automatically collect through cookies or other means when you use our services, including:

A.设备或软件信息，例如您的移动设备、网页浏览器或您用于接入我们的服务的其他程序所提供的配置信息、您的IP地址和您的移动设备所用的版本和设备识别码。

A. Device or software information, such as the configuration information provided by your mobile device, web browser, or other programs used to access our services, your IP address, and the version and device identification code of your mobile device.

B.您在使用我们服务时搜索和浏览的信息，例如您使用的搜索关键字、访问页面，以及您在使用我们服务时浏览或要求提供的其他信息。

B. Information you search and browse when using our services, such as the search keywords you use, accessed pages, and other information you browse or request to provide when using our services.

Figure 1: Excerpts for the original privacy policy.

Table 2: Basic statistical information of the interviewees

| Characteristic | Attribute | Subtotal | Percentage |
|---|---|---|---|
| Gender | Male | 42 | 68.85% |
| | Female | 19 | 31.15% |
| Age | < 18 | 1 | 1.64% |
| | 18-25 | 46 | 75.41% |
| | 26-30 | 12 | 19.67% |
| | 31-40 | 2 | 3.28% |
| | >40 | 0 | 0% |
| Education | Associate & below | 0 | 0% |
| | Bachelor's | 23 | 37.7% |
| | Graduate & above | 38 | 62.3% |

## A.5 Dataset documentation and intended uses

The following questions are copied from "Datasheets for Datasets" [1].

### A.5.1 Motivation

- **For what purpose was the dataset created?** (Was there a specific task in mind? Was there a specific gap that needed to be filled? Please provide a description.)

**Human:**

一、应用程序如何收集和使用个人信息。

Ⅰ. How Application collect and use personal information:

1、当你注册成为用户或使用服务时，需要您提供某些个人信息。注销账号时，这些信息将被将停止使用。

When you register as a user or use the service, you are required to provide certain personal information. This information will cease to be used when you cancel your account.

2、使用服务时，系统可能通过cookies或其他自动采集技术，采集设备或软件信息、搜索和浏览时产生的信息以及要求提供的其他信息。

When using the service, the system may collect device or software information, information generated during search and browsing, and other requested information through cookies or other automatic collection technologies.

Figure 2: Summarization of Human Expert.

To alleviate the common issues of "lengthy" and "incomprehensibility" that the public often encounters when reading privacy policies, and to avoid the risks of information security caused by over-authorization, we have carried out innovative work. While existing privacy policy corpus have somewhat alleviated the "lengthy" issue of reading privacy policies, they have not provided practical solutions for the "incomprehensibility" problem brought by complex sentence structures and professional vocabulary. Because of this, we have released the CAPP-130 corpus, which is a corpus consisting of 130 Chinese privacy policies and 38,889 sentences. In this corpus, we annotated 11 types of data practices and two special markers (used to identify the importance of privacy policies and potential privacy risks) and ultimately rewrote these policies into a language form that is easier to understand. This is different from other privacy policy summarization works, as CAPP-130 provides rewritten content, contributing to the development of generative natural language processing (NLP) tools in the field of privacy policies. Based on CAPP-130, we also conducted benchmark tests on the performance of classic deep-learning models in classification and generation tasks.

- **Who created this dataset (e.g., which team, research group) and on behalf of which entity (e.g., company, institution, organization)?**
  DIVER, an interdisciplinary research group at Zhejiang University.

- **Who funded the creation of the dataset?** (If there is an associated grant, please provide the name of the grantor and the grant name and number.)
  This work was supported in part by the National Key RD Program of China (2021YFB3101100) and NSFC grants (62102352, 62206207).

### A.5.2 Composition

- **What do the instances that comprise the dataset represent (e.g., documents, photos, people, countries)** (Are there multiple types of instances (e.g., movies, users, and ratings; people and interactions between them; nodes and edges)? Please provide a description.)
  The sentences in the privacy policy constitute instances of the CAPP-130, which include "Original Sentence", "Data Practice Category", "Special Marking", and the "Rewritten Sentence".

- **How many instances are there in total** (of each type, if appropriate)?
  The dataset contains 130 Chinese privacy policies of mainstream applications, including 38889 preprocessed sentences.

- **Does the dataset contain all possible instances or is it a sample (not necessarily random) of instances from a larger set?** (If the dataset is a sample, then what is the larger set? Is the sample representative of the larger set (e.g., geographic coverage)? If so, please describe how this representativeness was validated/verified. If it is not representative of the larger set, please describe why not (e.g., to cover a more diverse range of instances, because instances were withheld or unavailable).)
  This dataset is a sample from a larger set. Details could be checked in Section 3.1.

- **What data does each instance consist of?** ("Raw" data (e.g., unprocessed text or images)or features? In either case, please provide a description.)

Figure 3: Summarizations of ERNIE Bot and Tongyi Qianwen.

Each instance has "Original Sentence", "Sentence ID" and annotations, including "Data Practices Category", " Special Marking", and "Rewritten Sentence".

- **Is there a label or target associated with each instance?** If so, please provide a description.

  Privacy policy annotations consist of three parts: "Data Practices Category", "Special Marking", and "Rewritten Sentence". Details could be checked in Section 3.1 of the main paper.

- **Is any information missing from individual instances?** (If so, please provide a description, explaining why this information is missing (e.g., because it was unavailable). This does not include intentionally removed information, but might include, e.g., redacted text.)

  If a sentence lack "Special Marking", then they don't require rewriting. Since these sentences are some insignificant declarative documents, there is no need to summarize and rewrite them.

**TextRank:**

一、我们如何收集和使用您的个人信息

Ⅰ. How we collect and use your personal information:

个人信息是指以电子或者其他方式记录的能够单独或者与其他信息结合识别特定自然人身份或者反映特定自然人活动情况的各种信息

Personal information refers to various information recorded electronically or by other means that can identify a specific individual or reflect specific activities of an individual

1、您提供的信息

1. Information you provide:

您注册成为我们的用户或使用我们的服务时，向我们提供的相关个人信息，包括您的【姓名（名称）、性别、年龄、地址、电话号码、电子邮箱地址、身份证件号码、缴费相关的银行信息以及联系人信息等】。您提供的上述信息，将在您使用本网站服务期间持续授权我们使用。在您注销账号时，我们将停止使用上述信息。

When you register as our user or use our services, you provide us with relevant personal information, including your name, gender, age, address, phone number, email address, identification card number, bank information related to payments, and contact information. The aforementioned information you provide will be continuously authorized for our use during your use of this website's services. We will stop using the aforementioned information when you cancel your account.

2、您在使用服务过程中产生的信息

2. Information generated during your use of the service:

（1）日志信息

(1) Log information:

指您使用我们服务时，系统可能通过cookies或其他方式自动采集的技术信息，包括：

Refers to technical information that the system may automatically collect through cookies or other means when you use our services, including:

A.设备或软件信息，例如您的移动设备、网页浏览器或您用于接入我们的服务的其他程序所提供的配置信息、您的IP地址和您的移动设备所用的版本和设备识别码。

A. Device or software information, such as the configuration information provided by your mobile device, web browser, or other programs used to access our services, your IP address, and the version and device identification code of your mobile device.

B.您在使用我们服务时搜索和浏览的信息，例如您使用的搜索关键字、访问页面，以及您在使用我们服务时浏览或要求提供的其他信息。

B. Information you search and browse when using our services, such as the search keywords you use, accessed pages, and other information you browse or request to provide when using our services.

**Checkpoint:**

关于收集、存储和使用您的个人信息的说明

Statement on the Collection, Storage, and Use of Your Personal Information

Figure 4: Summarizations of TextRank and Checkpoint.

- **Are relationships between individual instances made explicit (e.g., users' movie ratings, social network links)?** ( If so, please describe how these relationships are made explicit.) Indeed, all instances have a unique identifier "id" that can be linked to the corresponding privacy policy.

- **Are there recommended data splits (e.g., training, development/validation, testing)?** (If so, please provide a description of these splits, explaining the rationale behind them.) Yes. We did a train-validation-test split on the dataset, see Section A.1.

- **Are there any errors, sources of noise, or redundancies in the dataset?** (If so, please provide a description.) The creation of the CAPP-130 corpus involved annotations by legal experts, which may introduce bias based on their individual interpretations and perspectives. While efforts have been made to ensure consistency, such as strict annotation standards to improve Cohen's kappa coefficient of the annotations made by legal experts, variations in interpretations may impact the quality of the corpus.

- **Is the dataset self-contained, or does it link to or otherwise rely on external resources (e.g., websites, tweets, other datasets)?**(If it links to or relies on external resources, a) are

there guarantees that they will exist, and remain constant, over time; b) are there official archival versions of the complete dataset (i.e., including the external resources as they existed at the time the dataset was created); c) are there any restrictions (e.g., licenses, fees) associated with any of the external resources that might apply to a future user? Please provide descriptions of all external resources and any restrictions associated with them, as well as links or other access points, as appropriate.)

The dataset in question does not link to or rely on external resources. It is a self-contained dataset.

- **Does the dataset contain data that might be considered confidential (e.g., data that is protected by legal privilege or by doctor-patient confidentiality, data that includes the content of individuals' non-public communications)?** (If so, please provide a description.)

  No, all these privacy policies are publicly available.

- **Does the dataset contain data that, if viewed directly, might be offensive, insulting, threatening, or might otherwise cause anxiety?** (If so, please describe why.)

  No, CAPP-130 does not contain data that might be offensive, insulting, threatening, or cause anxiety because the privacy policy is a public agreement.

- **Does the dataset relate to people?** (If not, you may skip the remaining questions in this section.)

  No, the dataset does not relate to people.

- **Does the dataset identify any subpopulations (e.g., by age, gender)?** If so, please describe how these subpopulations are identified and provide a description of their respective distributions within the dataset.

  N/A.

- **Is it possible to identify individuals (i.e., one or more natural persons), either directly or indirectly (i.e., in combination with other data) from the dataset?**(If so, please describe how how these subpopulations are identified and provide a description of their respective distributions within the dataset.)

  N/A.

- **Does the dataset contain data that might be considered sensitive in any way (e.g., data that reveals racial or ethnic origins, sexual orientations, religious beliefs, political opinions or union memberships, or locations; financial or health data; biometric or genetic data; forms of government identification, such as social security numbers; criminal history)?** (If so, please provide a description.)

  N/A.

- **Any other comments?**

  None.

### A.5.3 Collection Process

- **How was the data associated with each instance acquired?** (Was the data directly observable (e.g., raw text, movie ratings), reported by subjects (e.g., survey responses), or indirectly inferred/derived from other data (e.g., part-of-speech tags, model-based guesses for age or language)? If data was reported by subjects or indirectly inferred/derived from other data, was the data validated/verified? If so, please describe how.)

  Please check Section 3.1 in the main paper.

- **What mechanisms or procedures were used to collect the data (e.g., hardware apparatus or sensor, manual human curation, software program, software API)?** (How were these mechanisms or procedures validated?)

  We wrote a simple web crawler program and downloaded these public privacy policies from Xiaomi App Store and Huawei App Store.

- **If the dataset is a sample from a larger set, what was the sampling strategy (e.g., deterministic, probabilistic with specific sampling probabilities)?**

  All privacy policies are collected through the official links provided by Xiaomi App Store and Huawei App Store. We use web crawling technology to download the privacy policies

of the top 100 Chinese apps according to the download rankings of each app store. Then, we remove the overlapping privacy policies and check the completeness of the content, resulting in 130 usable privacy policies. Finally, we use regular expressions to divide the collected privacy policies into sentence-level samples.

- **Who was involved in the data collection process (e.g., students, crowdworkers, contractors) and how were they compensated (e.g., how much were crowdworkers paid)?**
  This dataset was voluntarily annotated by the authors and members of the DIVER group.

- **Over what timeframe was the data collected? (Does this timeframe match the creation timeframe of the data associated with the instances (e.g., recent crawl of old news articles)?** If not, please describe the timeframe in which the data associated with the instances was created.)
  We collect privacy policy updates until December 2022.

- **Were any ethical review processes conducted (e.g., by an institutional review board)?** (If so, please provide a description of these review processes, including the outcomes, as well as a link or other access point to any supporting documentation.)
  Yes. We have conducted an internal ethical review process by the Zhejiang University ethical team.

- **Does the dataset relate to people?** (If not, you may skip the remaining questions in this section.)
  No, the dataset does not relate to people.

- **Did you collect the data from the individuals in question directly, or obtain it via third parties or other sources (e.g., websites)?** Were the individuals in question notified about the data collection? If so, please describe(or show with screenshots or other information) how notice was provided, and provide a link or other access point to, or otherwise reproduce, the exact language of the notification itself.
  N/A.

- **Were the individuals in question notified about the data collection?** (If so, please describe (or show with screenshots or other information) how notice was provided, and provide a link or other access point to, or otherwise reproduce, the exact language of the notification itself.)
  N/A.

- **Did the individuals in question consent to the collection and use of their data?** (If so, please describe (or show with screenshots or other information) how consent was requested and provided, and provide a link or other access point to, or otherwise reproduce, the exact language to which the individuals consented.)
  N/A.

- **If consent was obtained, were the consenting individuals provided with a mechanism to revoke their consent in the future or for certain uses?** (If so, please provide a description, as well as a link or other access point to the mechanism (if appropriate).)
  N/A.

- **Has an analysis of the potential impact of the dataset and its use on data subjects (e.g., a data protection impact analysis) been conducted?** (If so, please provide a description of this analysis, including the outcomes, as well as a link or other access point to any supporting documentation.)
  N/A.

- **Any other comments?**
  None.

### A.5.4  Preprocessing/cleaning/labeling

- **Was any preprocessing/cleaning/labeling of the data done (e.g., discretization or bucketing, tokenization, part-of-speech tagging, SIFT feature extraction, removal of instances, processing of missing values)?** (If so, please provide a description. If not, you may skip the remainder of the questions in this section.)
  We removed the HTML tags because they are meaningless for privacy policy research.

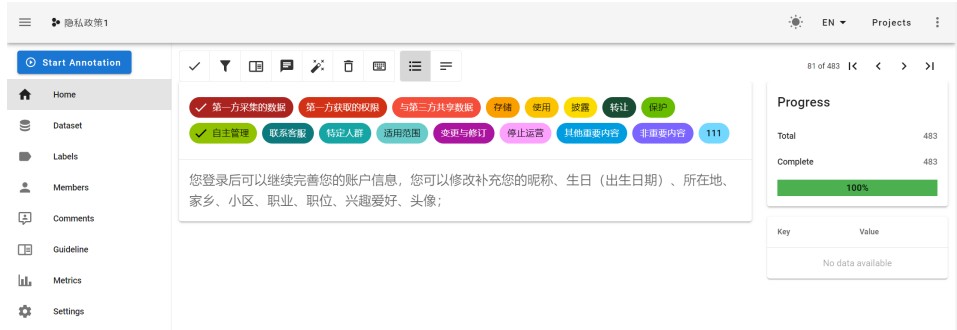

Figure 5: Annotation of Privacy Policy in Doccano

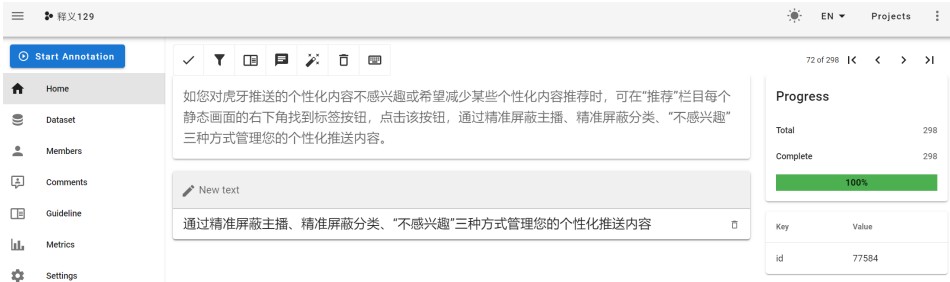

Figure 6: Rewritten of Privacy Policy in Doccano

- **Was the "raw" data saved in addition to the preprocessed/cleaned/labeled data (e.g., to support unanticipated future uses)?** (If so, please provide a link or other access point to the "raw" data.)

  yes, saved "raw" data in GitHub.

- **Is the software used to preprocess/clean/label the instances available?** (If so, please provide a link or other access point.)

  Yes. All the annotation work is done using the Doccano annotation platform by [2]. The link of Doccano is as follows: *https://github.com/doccano/doccano*. Screenshots are shown in Figure 5 and Figure 6. Our data, annotation guidelines, and related code are publicly available (*https://github.com/EnlightenedAI/CAPP-130*) to ensure reproducibility.

- **Any other comments?**

  None.

### A.5.5 Uses

- **Has the dataset been used for any tasks already?** (If so, please provide a description.)

  CAPP-130 is a new dataset of summarized and rewritten Chinese privacy policies. We benchmark the performance of classical machine learning through extensive experiments, building models, and state-of-the-art deep learning models on automatic text summarization, text generation, and text classification. The experimental results are detailed in Section 4 of the main paper.

- **Is there a repository that links to any or all papers or systems that use the dataset?** (If so, please provide a link or other access point.)

  Our data, annotation guidelines, benchmark models, and source code are publicly available (*https://github.com/EnlightenedAI/CAPP-130*) to ensure reproducibility.

- **What (other) tasks could the dataset be used for?**

  CAPP-130 can be used for Chinese text generation, text classification, and automatic text summarization. We provided application examples at *https://github.com/EnlightenedAI/CAPP-130*.

- **Is there anything about the composition of the dataset or the way it was collected and preprocessed/cleaned/labeled that might impact future uses?** (For example, is there anything that a future user might need to know to avoid uses that could result in unfair treatment of individuals or groups (e.g., stereotyping, quality of service issues) or other undesirable harms (e.g., financial harms, legal risks) If so, please provide a description. Is there anything a future user could do to mitigate these undesirable harms?)

  No.

- **Are there tasks for which the dataset should not be used?** (If so, please provide a description.)

  No.

- **Any other comments?**

  None.

### A.5.6   Distribution

- **Will the dataset be distributed to third parties outside of the entity (e.g., company, institution, organization) on behalf of which the dataset was created?** (If so, please provide a description.)

  No.

- **How will the dataset will be distributed (e.g., tarball on website, API, GitHub)?** (Does the dataset have a digital object identifier (DOI)?)

  It is released on Github at $https://github.com/EnlightenedAI/CAPP\text{-}130$. No DOI.

- **When will the dataset be distributed?**

  Before the conference.

- **Will the dataset be distributed under a copyright or other intellectual property (IP) license, and/or under applicable terms of use (ToU)?** (If so, please describe this license and/or ToU, and provide a link or other access point to, or otherwise reproduce, any relevant licensing terms or ToU, as well as any fees associated with these restrictions.)

  Apache License 2.0. ($https://github.com/YLab\text{-}Open/METS\text{-}CoV/blob/main/LICENSE$)

- **Have any third parties imposed IP-based or other restrictions on the data associated with the instances?** (If so, please describe these restrictions, and provide a link or other access point to, or otherwise reproduce, any relevant licensing terms, as well as any fees associated with these restrictions.)

  No.

- **Do any export controls or other regulatory restrictions apply to the dataset or to individual instances?** (If so, please describe these restrictions, and provide a link or other access point to, or otherwise reproduce, any supporting documentation.)

  No.

- **Any other comments?**

  None.

### A.5.7   Maintenance

- **Who is supporting/hosting/maintaining the dataset?**

  DIVER, an interdisciplinary research group at Zhejiang University.

- **How can the owner/curator/manager of the dataset be contacted (e.g., email address)?**

  E-mail addresses are at the top of this document.

- **Is there an erratum?** (If so, please provide a link or other access point.)

  No.

- **Will the dataset be updated (e.g., to correct labeling errors, add new instances, delete instances')?** (If so, please describe how often, by whom, and how updates will be communicated to users (e.g., mailing list, GitHub)?)

No. If we plan to update the dataset in the future, we will elaborate on the reason on our GitHub repository.

- **If the dataset relates to people, are there applicable limits on the retention of the data associated with the instances (e.g., were individuals in question told that their data would be retained for a fixed period of time and then deleted)?** (If so, please describe these limits and explain how they will be enforced.)

  No.

- **Will older versions of the dataset continue to be supported/hosted/maintained?** (If so, please describe how. If not, please describe how its obsolescence will be communicated to users.)

  Yes. If we plan to update the data, we will maintain the old version and then release the follow-up version, for example, CAPP-130-V2.0.

- **If others want to extend/augment/build on/contribute to the dataset, is there a mechanism for them to do so?** (If so, please provide a description. Will these contributions be validated/verified? If so, please describe how. If not, why not? Is there a process for communicating/distributing these contributions to other users? If so, please provide a description.)

  Yes. For data annotation, researchers could carefully check our annotation guidelines in Git Hub to ensure consistency. And if others want to contribute to the dataset, they could submit a pull request or contact us via email.

- **Any other comments?**

  None.

## A.6  Accessibility

1. Links to access the dataset and its metadata. (`https://github.com/EnlightenedAI/CAPP-130`)

2. The data is saved in a JSON format, where an example is shown in the README.md file.

3. DIVER research group will maintain this dataset on the official Github (`https://github.com/EnlightenedAI/CAPP-130`) account.

4. Apache License 2.0. (`https://github.com/YLab-Open/METS-CoV/blob/main/LICENSE`)

## A.7  Data Usage

The authors bear all responsibility in case of violation of rights.