# OpenReview forum: "CAPP-130: A Corpus of Chinese Application Privacy Policy Summarization and Interpretation"
_NeurIPS.cc/2023/Track/Datasets_and_Benchmarks — NeurIPS 2023 Datasets and Benchmarks Poster_

### Official Review · Reviewer_nRQo · 2023-06-28
**A valuable and soundly curated complement to the few, and mostly US-centric, high-quality privacy policy annotation datasets -- but appendix missing**

**Rating:** 8
**Confidence:** 4
**Clarity:** The paper is well-written, well-motiv…

**Strengths:**

- The selection and collection process for the policies covered is legit and sound. The annotations were done by different legal experts and annotations of the same policy by different experts were cross-validated with high Cohen's kappa values. Noteworthily, many other annotation efforts lack these cross-checks, making the work at hand stand out in matters of solidity.

- The cursory statistics on the resulting dataset presented in 3.1 complete this impression. Given the labor-intensity of properly annotating privacy policies and the fact that most comparable work therefore uses the outdated and US-focused OPP-115, the value of the annotated corpus is definitely higher than one might possibly assume.

- The summarization framework also follows a reasonable and straightforward approach. The variety of used extraction and generative transformation frameworks used is much appreciated.

**Additional Feedback:**

Assuming the appendix actually confirms the statements being made, the paper is solid, sound, and valuable. Besides providing a valuable complement to the few, and mostly US-centric, high-quality privacy policy annotation datasets, it also provides an interesting and novel approach to (semi-) automatically generate more readable privacy policies from existing ones. Transferring this approach to non-chinese policies would be an interesting endeavor. If the authors find the time, I would appreciate additionally including SBERT in the evaluation.

Some minor things

* the abstract should (and easily could) be shortened.
* I feel the language in privacy policies is not only technical but also (and probably more importantly) legalese. This could also be included in the argumentation.
* sometimes, blanks are missing around sentence ends and citations.
* line 56: "focus is on extracting private information from privacy policies" - I don't understand this.
* line 245: EREIE --> ERNIE

**Correctness:**

yes, everything appears o be correct - as far as it can be assessed wothout the missing appendix

**Documentation:**

The appendix is hardly missing

**Ethics:**

no concerns

**Limitations:**

yes

**Opportunities For Improvement:**

- Given that the approach works on the sentence level, however, I wonder why the authors didn't include Sentence-BERT (which is also available in Chinese) in their experiments. I would expect even better results here. The detailed experiment results are, however, not available for review due to a missing appendix (see below).

- Unfortunately, the code and the dataset are not available for inspection/review, even though the authors provide a github link. It is thus hard to assess the details behind the presented work.

- The most important drawback of the paper is that the appendix is missing. In the text, the authors refer to this appendix not only regarding "the originally extracted excerpts as well as summary results" but also the detailed experiment parameters and results (e.g., "Textrank considers nearly every sentence important"). Especially the detailed experiment results would have been important for getting a deeper understanding and insight. Lastly, the appendix is also referred to regarding the training details in checklist question 3 b) and d). This appendix would have been important for properly assesing the paper at hand. The authors should add this missing appendix (and an actually filled repo) asap.

**Relation To Prior Work:**

Relevant literature is basically well-covered. However, another strand of possibly relevant related work (especially for the first part of section 3) are formal privacy / transparency languages such as LPL (layered privacy language), TILT (transparency information language and toolkit), PPL (PrimeLife policy language), or even P3P (platform for privacy preferences) - the motivation behind these is typically - amongst other things - also to make policies more readable, comparable, etc.

**Summary And Contributions:**

This paper presents CAPP-130, an annotated chinese privacy policy corpus comprising 130 prominent apps, and TCSI-pp, a method for summarizing such policies based on the annotated corpus. The corpus is annotated by legal experts and the summarization process employs a combination of automated information extraction and, on that basis, transformer-based text generation to produce rewritten policies that should be easier to comprehend. As for the extraction and the generative transforation, different frameworks and combinations thereof are experimentally evaluated.

---

> ### Author Response · Authors · 2023-08-21
>
> Thank you for your insightful and encouraging comments. We carefully and thoroughly revised the manuscript to address all the comments. The main changes are highlighted in blue and the detailed point-to-point responses are provided as follows.
>
> **Question 1:**
> Given that the approach works on the sentence level, however, I wonder why the authors didn't include Sentence-BERT (which is also available in Chinese) in their experiments. I would expect even better results here.
>
> **Response to Question 1:**
> Thank you for your valuable suggestions. we incorporated SBERT in the revision.  The experimental results are shown in Table 1. We added the corresponding discussion in Section 4.2.
>
> Table 1: Experimental results on the multi-classification and binary classification.
> |Model| Topic-multi Micro-F1 |Topic-multi Macro-F1|Important-binary Micro-F1 |Important-binary Macro-F1 |Risk-binary Micro-F1 |Risk-binary Macro-F1|
> |----|----|----|----|----|----|----|
> |RoBERTa|**0.819**|**0.841**|**0.897**|**0.899**|0.920|0.711|
> |Bert	|0.802	 |0.820	| 0.895    |0.896   |0.921	|0.719 |
> |mBERT	|0.809	 |0.821    |0.889    |0.889    |0.918 	|0.709|
> |SBERT	|0.781  |0.794 	| 0.875    |0.874    |0.917	|0.689|
> |PERT	|0.801	 |0.812	| 0.895   |0.897    |**0.922**|**0.716**|
> |ERNIE	|0.807	 |0.821	| 0.895    |0.896    |0.921| 0.702|
>
> **Question 2:**
> The detailed experiment results are, however, not available for review due to a missing appendix (see below).
>
> **Response to Question 2:**
> We incorporated the appendix in the revision.
>
>
> **Question 3:**
> Unfortunately, the code and the dataset are not available for inspection/review, even though the authors provide a github link. It is thus hard to assess the details behind the presented work.
>
> **Response to Question 3:**
> We made our Github repository public, you can access it by https://github.com/EnlightenedAI/CAPP-130.
>
> **Question 4:**
> The most important drawback of the paper is that the appendix is missing. In the text, the authors refer to this appendix not only regarding ''the originally extracted excerpts as well as summary results'' but also the detailed experiment parameters and results (e.g., ''Textrank considers nearly every sentence important''). Especially the detailed experiment results would have been important for getting a deeper understanding and insight. Lastly, the appendix is also referred to regarding the training details in checklist question 3 b) and d). This appendix would have been important for properly assesing the paper at hand. The authors should add this missing appendix (and an actually filled repo) asap.
>
> **Response to Question 4:**
> We incorporated the appendix in the revision.
>
> **Relation To Prior Work:**
> Relevant literature is basically well-covered. However, another strand of possibly relevant related work (especially for the first part of section 3) are formal privacy / transparency languages such as LPL (layered privacy language), TILT (transparency information language and toolkit), PPL (PrimeLife policy language), or even P3P (platform for privacy preferences) - the motivation behind these is typically - amongst other things - also to make policies more readable, comparable, etc.
>
> **Response to ''Relation To Prior Work'':**
> Following your valuable suggestions, we incorporated LPL, TILT, PPL, and P3P into the revision, and added the corresponding discussion in Section 1 and Section 2.
>
> **Additional Feedback:**
> If the authors find the time, I would appreciate additionally including SBERT in the evaluation.
>
> Some minor things
>
> the abstract should (and easily could) be shortened.
> I feel the language in privacy policies is not only technical but also (and probably more importantly) legalese. This could also be included in the argumentation.
> sometimes, blanks are missing around sentence ends and citations.
> line 56: ''focus is on extracting private information from privacy policies'' - I don't understand this.
> line 245: EREIE --> ERNIE
>
> **Response to ''Additional Feedback'':**
> We appreciate your careful comments and made the following modifications:
> (1) We incorporated SBERT into section 4.2;
> (2) We shorten the abstract;
> (3) We revised ''technical language'' to ''technobabble and legalese'';
> (4) We revised ''focus is on extracting private information from privacy policies'' to ''typically focus on extracting sentences including private information related to certain topics from privacy policies'';
> (5) We revised ''EREIE'' to ''ERNIE''.
>
> Please let us know if these additions are satisfactory. Thank you for taking the time to review our paper and providing such valuable feedback.

---

> > ### Comment · Reviewer_nRQo · 2023-08-28
> > **thanks for the changes**
> >
> > ... quite interesting that SBERT doesn't perform better.
> >
> > Let me reinforce the aspect that (finally) having a substantial, carefully annotated privacy policy dataset in a non-english language is a *really* valuable contribution. It's incredible how much research basically refers to the same two or three english datasets here. This makes the paper way more fitting to the track than the other ones I reviewed.

---

### Official Review · Reviewer_b2mX · 2023-07-21
**see the summary and contributions**

**Rating:** 3
**Confidence:** 5

**Strengths:**

The problem of summarizing privacy policies is interesting and practical.
Therefore, the introduction of a novel dataset allows the research community to foster the development and automation of this real-world application.


**Additional Feedback:**

no additional feedback

**Clarity:**

The paper needs some revisions because of numerous typo and presentation problems:
EREIE -> ERNIE (L 245)
Textrank -> TextRank (L 307)
“Topic”, -> “Topic,” (L 136)
The quality of Figure 2 is low and it can be improved.
Figures and Tables should be positioned on top or bottom of the page.
20630 -> 20,630 (L 61)
Natural Language Processing -> NLP (L 78)
“data practice categorie” -> “data practice categories” (L 91)


**Correctness:**

it is difficult to evaluate the correctness of the proposed dataset because the annotation process among annotators is not discussed.

**Documentation:**

There is no sufficient documentation of the proposed dataset. For instance, there I no information about the availability and maintenance.


**Ethics:**

No, there are no ethics concerns

**Limitations:**

The authors did not addressed adequately the limitations of their work. The only sentence about it is the following (L 328-329): “However, TCSI-pp still faces issues such as insufficient intuitiveness and a low level of customization.” However the authors never conducted experiments to demonstrate quantitatively and qualitatively such a limitation.
See “Opportunities For Improvement.”

**Opportunities For Improvement:**

The proposed work has several limitations that make the paper very weak and not ready to be published.
It is not mentioned how the annotation process has been conducted. There is a 0.907 of inter-annotator agreement. Yet, how the data has been split among annotators to make an agreement (I think the data should be overlapped)? Further, if there is data overlapping among annotators, how is the process of data labelling (majority vote rule)?
There is no information about the data splitting into train-validation-test.
The baselines are too poor. For example, the mBART model [1] has not be considered, although it supports Chinese. Why did not you considered RoBERTa (i.e., XLM-RoBERTa [2] that supports Chinese) instead of BERT?
Why did not you utilize the commonly used micro-F1 e macro-F1 metrics for classification tasks?
Why did you use ERNIE for binary classification and BERT for multi-class classification? I see that ERNIE is better for the binary task, but this should be explored on the validation set (which is never mentioned in the paper).
The proposed pipeline is not novel. Indeed, it is a well-known extract-then-abstract approach. Why did not you compared with seq-to-seq approaches (i.e., without extracting only relevant sentences with a classifier)?
The evaluation metrics for the rewritten sentences—based on a syntactic assessment with ROUGE—are not so robust. You should also use semantic-based metrics such as BERTScore [3] and factuality-aware metrics such as BARTScore [4]. Furthermore, you should also consider novel carbon-aware metrics such as Carburacy [5].

Other comments:
The citations should not be in plain text.
You have never cited ROUGE.
Why ChatGPT is GPT-4? They are different.
The GitHub repository is empty.

References:
[1] Liu, Yinhan, et al. "Multilingual denoising pre-training for neural machine translation." Transactions of the Association for Computational Linguistics 8 (2020): 726-742.
[2] Conneau, Alexis, et al. "Unsupervised cross-lingual representation learning at scale." arXiv preprint arXiv:1911.02116 (2019).
[3] Zhang, Tianyi, et al. "Bertscore: Evaluating text generation with bert." arXiv preprint arXiv:1904.09675 (2019).
[4] Yuan, Weizhe, Graham Neubig, and Pengfei Liu. "Bartscore: Evaluating generated text as text generation." Advances in Neural Information Processing Systems 34 (2021): 27263-27277.
[5] Moro, Gianluca, Luca Ragazzi, and Lorenzo Valgimigli. "Carburacy: summarization models tuning and comparison in eco-sustainable regimes with a novel carbon-aware accuracy." Proceedings of the AAAI Conference on Artificial Intelligence. Vol. 37. No. 12. 2023.

**Relation To Prior Work:**

The related work is clearly discussed.

**Summary And Contributions:**

The work focuses on text summarization for Chinese privacy policies. Such documents are complex to read and comprehend because of their length and the technical language, so summarization techniques can be helpful to let users quickly grasp the meaningful information. Since no datasets exist for this task, the authors present CAPP-130, the first corpus for text summarization of Chinese privacy policies. They also propose TCSI-pp to generate abstractive summaries of privacy policies, and they compare their solution with large language models, showing better performance.

---

> ### Author Response · Authors · 2023-08-21
>
> Thank you for your careful and encouraging comments. We carefully and thoroughly revised the manuscript to address all the comments. The main changes are highlighted in blue and the detailed point-to-point responses are provided as follows.
>
> **Question 1:**
> It is not mentioned how the annotation process has been conducted. There is a 0.907 of inter-annotator agreement. Yet, how the data has been split among annotators to make an agreement (I think the data should be overlapped)? Further, if there is data overlapping among annotators, how is the process of data labelling (majority vote rule)?
>
> **Response to Question 1:**
> Thank you for your valuable suggestions. In the annotation process, we randomly select 20 privacy policies, which are jointly annotated by three legal experts, resulting in a Cohen Kappa coefficient of 0.907. This result strongly validates the annotation reliability of each legal expert. On the basis of high reliability, the remaining privacy policies are annotated by one of the legal experts considering the cost performance. For the 20 selected privacy policies annotated by three legal experts, we employ the majority vote as you mentioned. In the revision, we clarified the misunderstanding in Section 3.1.
>
>
> **Question 2:**
> There is no information about the data splitting into train-validation-test.
>
> **Response to Question 2:**
> Thank you for your valuable suggestions. We randomly split the classification corpus into training data, validation data, and testing data at a ratio of 70:15:15. The rewritten corpus is randomly split into training data, validation data, and testing data at a ratio of 80:10:10. The details of each corpus are shown in Table 1. We added the corresponding discussion in Appendix A.1.
>
> Table 1: The statistics of train-validation-test data.
> |corpus| #all samples |#training samples  |#validation samples  |#testing samples|
> |----|----|----|----|----|
> |important-binary |38989	|27222 |5883	|5884 |
> |risk-binary|20505|14338|3083 |3084 |
> |topic-multi |20268 |14190 |3043 | 3035 |
> |rewritten	|19570|15656	|1957   |1957|
>
> **Question 3:**
> The baselines are too poor. For example, the mBART model has not be considered, although it supports Chinese. Why did not you considered RoBERTa (i.e., XLM-RoBERTa that supports Chinese) instead of BERT? Why did not you utilize the commonly used micro-F1 e macro-F1 metrics for classification tasks? Why did you use ERNIE for binary classification and BERT for multi-class classification? I see that ERNIE is better for the binary task, but this should be explored on the validation set (which is never mentioned in the paper).
>
>
> **Response to Question 3:**
> Following your valuable suggestions, we incorporated mBART and RoBERTa in the revision. Furthermore, we added Pert[1] and SBERT as classification benchmarks. We also changed to using micro-F1 and macro-F1 as performance metrics. The experimental results show that RoBERTa is superior to other models on the classification tasks including both multi-classification and binary classification as shown in Table 2. Therefore, we used RoBERTa as the classification model when implementing the summarization tool TCSI-pp-zh in the revision. We added the corresponding discussion in Section 4.2.
>
> [1] Cui Y, Yang Z, Liu T. PERT: pre-training BERT with permuted language model[J]. arXiv preprint arXiv:2203.06906, 2022.
>
> Table 2: Experimental results on the multi-classification and binary classification.
> |Model| topic-multi Micro-F1 |topic-multi Macro-F1|important-binary Micro-F1 |important-binary Macro-F1 |risk-binary Micro-F1 |risk-binary Macro-F1|
> |----|----|----|----|----|----|----|
> |RoBERTa|**0.819**|**0.841**|**0.897**|**0.899**|0.920|0.711|
> |Bert	|0.802	 |0.820	| 0.895    |0.896   |0.921	|0.719 |
> |mBERT	|0.809	 |0.821    |0.889    |0.889    |0.918 	|0.709|
> |SBERT	|0.781  |0.794 	| 0.875    |0.874    |0.917	|0.689|
> |PERT	|0.801	 |0.812	| 0.895   |0.897    |**0.922**|**0.716**|
> |ERNIE	|0.807	 |0.821	| 0.895    |0.896    |0.921| 0.702|
>
> **Question 4:** The proposed pipeline is not novel. Indeed, it is a well-known extract-then-abstract approach. Why did not you compared with seq-to-seq approaches (i.e., without extracting only relevant sentences with a classifier)?
>
> **Response to Question 4:**
> In the submitted paper, we have compared with the seq-to-seq approaches including GPT-4, ERNIEbot, Tongyi QiWen, and Checkpoint (Randeng-Pegasus-523M-Summary-Chinese), and presented the results in Figure 3. To make it clearer, we presented more experimental results in Figures 3,4 in the appendix.

---

> ### Author Response · Authors · 2023-08-21
>
> **Question 5:**
> The evaluation metrics for the rewritten sentences—based on a syntactic assessment with ROUGE—are not so robust. You should also use semantic-based metrics such as BERTScore  and factuality-aware metrics such as BARTScore. Furthermore, you should also consider novel carbon-aware metrics such as Carburacy.
>
>
> **Response to Question 5:**
> Following your valuable suggestions, we incorporated  BARTScore, BERTScore, and Carburacy in the revision. The experimental results are shown in Table 3. We added the corresponding discussion in Section 4.3.
>
> Table 3: Experimental results on the generative models.
> | Model|  ROUGE-1 | 	ROUGE-2  | 	ROUGE-L|  Bert-score |  Bart-score | Carburacy |
> |----|----|----|----|----|----|----|
> |mT5|**0.753**|**0.609**|**0.733**|**0.888**|**-4.577**|**0.833**|
> |Bert2gpt|0.751|0.574|0.720|0.872|-4.964|0.764|
> |RoBerta2gpt|0.749	|0.577|0.719 |0.872|-4.975 |0.755|
> |Bert2bert|0.718|0.535|0.689 |0.864 |-5.020 |0.747|
> |ERNIE2gpt|0.623|0.406|0.581 |0.809|-5.716|0.715|
>
> **Question 6:**
> Other comments: The citations should not be in plain text. You have never cited ROUGE. Why ChatGPT is GPT-4? They are different. The GitHub repository is empty.
>
> **Response to Question 6:**
> We appreciate your careful comments and made the following modifications:
> (1) We changed the bibliography style from plain text to abbrv;
> (2) We added the reference to ROUGE;
> (3) We revised ChatGPT(GPT-4) to GPT-4;
> (4) We made our Github repository public, you can access it by https://github.com/EnlightenedAI/CAPP-130.
>
> **Limitations:**
> The authors did not address adequately the limitations of their work. The only sentence about it is the following (L 328-329): ''However, TCSI-pp still faces issues such as insufficient intuitiveness and a low level of customization.'' However, the authors never conducted experiments to demonstrate quantitatively and qualitatively such a limitation.
>
> **Response to Limitations:**
> Quantifying or qualifying ''intuitiveness'' may be challenging or even unfeasible, as it is a subjective experience. We are in the process of designing a dashboard that will incorporate visual elements like icons and symbols, aiming to align summarization more closely with users' intuitive perceptions. While users can select specific content from the 11 data practices available in our model, they are unable to access information beyond these 11 data practice categories. In the future, we intend to explore additional data practice categories to enhance the ``customization'' of our summarization tool. We added the corresponding discussion in Section 5.
>
> **Clarity:** The paper needs some revisions because of numerous typo and presentation problems: EREIE -> ERNIE (L 245) Textrank -> TextRank (L 307) ''Topic'', -> ''Topic,'' (L 136) The quality of Figure 2 is low and it can be improved. Figures and Tables should be positioned on top or bottom of the page. 20630 -> 20,630 (L 61) Natural Language Processing -> NLP (L 78) “data practice categorie” -> “data practice categories” (L 91)
>
> **Response to Clarity:**
> Thank you for your careful comments. We have fixed them in the revision.
>
> **Documentation:**
> There is no sufficient documentation of the proposed dataset. For instance, there I no information about the availability and maintenance.
>
> **Response to Documentation:**
> Following your valuable suggestions, we added the ''Datasheets for Datasets'' in Appendix A.5 and made our Github repository public. You can access it by https://github.com/EnlightenedAI/CAPP-130.
>
> Thank you for devoting the time to review our paper and for offering such insightful suggestions. Please let us know if these additions meet your expectations.

---

> ### Author Response · Authors · 2023-08-31
>
> Thank you for taking the time to review our paper and providing such valuable feedback. Please let us know if these additions are satisfactory.

---

### Official Review · Reviewer_X4HM · 2023-07-22
**Useful corpus with sufficient scale in privacy policy study domain**

**Rating:** 6
**Confidence:** 4
**Correctness:** The motivation and flow of the study …

**Strengths:**

- This study provides a valuable resource in privacy policy domain (for Chinese applications), suggesting detailed collection and annotation scheme
- The created corpus is tested with publicly available language models and the size is sufficiently large
- The corpus is refined by law experts

**Additional Feedback:**

I wonder authors considered looking up privacy policies in Chinese law instead of ones in app stores, since I was first confused that the paper would handle privacy policy that applies like GDPR in European regions

**Clarity:**

The paper is well-written except some typos
- Ln 118 no spacing
- Table 3 some lowercase first letters

**Documentation:**

The construction process is well documented, but adopting law experts is itself a big human factor that might deter the replication of the study in other language or domain. I hope there be a guideline for reducing the variance coming from such human factors.

**Ethics:**

Only concern I have is the availability of the source data (document from the app stores of the private company) and hope authors checked the usability thoroughly

**Limitations:**

- In the sense of probable application, I think the related work section lacks study on privay policy itself and how related areas utilize such data
- Besides the lack of future application above, I think the paper needs to discuss how the proposed method and corpus contains the limitations, such as it being limited to the study on Chinese privacy policy, relying on machines for data collection, human factor in accommodating law experts in annotation and rewriting.


**Opportunities For Improvement:**

- Authors could provide more examples of the application of the created corpus, since the paper concentrates on the collection and annotation prodess of the corpus but lacks how it will be used in future by either researchers or policy makers

**Relation To Prior Work:**

As stated above, I hope there be more related work in current status and future application of privacy policy study

**Summary And Contributions:**

This study comprehensively studies privacy policies in Chinese, especially in terms of summarization technique and following corpus generation scheme. As a result, authors propose TCSI-pp (topic-controlled method for privacy policy summarization and interpretation framework) and CAPP-130 (Chinese application privacy policy corpus), and demonstrate the properties of the created corpus with available PLMs, LLMs, and other baselines. The corpus was annotated and refined by law experts, and contains about 50K annotations and 20K rewritten clauses.

---

> ### Author Response · Authors · 2023-08-21
>
> Thank you for your insightful and encouraging comments. We carefully and thoroughly revised the manuscript to address all the comments. The main changes are highlighted in blue and the detailed point-to-point responses are provided as follows.
>
> **Suggestion 1:**
> Authors could provide more examples of the application of the created corpus, since the paper concentrates on the collection and annotation prodess of the corpus but lacks how it will be used in future by either researchers or policy makers.
>
> **Response to Suggestion 1:**
> Thank you for your valuable suggestions. The CAPP-130 corpus can be used in areas such as text classification, text generation, and automatic text summarization. We added more examples of the application of CAPP-130 and presented the potential applications for researchers or policy makers in Appendix A.5.5.
>
> **Limitations 1:**
> In the sense of probable application, I think the related work section lacks study on privay policy itself and how related areas utilize such data.
>
> **Response to Limitations 1:**
> Following your valuable suggestions, we added the study on privacy policies and how related areas utilize privacy policies in Section 1 and Section 2.
>
> **Limitations 2:**
> Besides the lack of future application above, I think the paper needs to discuss how the proposed method and corpus contains the limitations, such as it being limited to the study on Chinese privacy policy, relying on machines for data collection, human factor in accommodating law experts in annotation and rewriting.
>
> **Response to Limitations 2:**
> Thank you for your valuable suggestions. We incorporated the limitations you mentioned in the revision. Our work is limited on Chinese privacy policies due to the limited time and funding of the past year. We have already started working on the English privacy polices including more than 100 English application privacy policies. The data collection of CAPP-130 indeed rely on machines, we will involve more human factors in collecting privacy policies for the future version. The creation of the CAPP-130 corpus involved annotations by legal experts, which may introduce bias based on their individual interpretations and perspectives. While efforts have been made to ensure consistency, such as strict annotation standards to improve Cohen's kappa coefficient of the annotations made by legal experts, variations in interpretations may impact the quality of the corpus. We will explore more methods to ensure consistency across legal experts. We added the corresponding discussion in Section 5 and Section A.5.2.
>
> **Clarity:**
> The paper is well-written except some typos
> Ln 118 no spacing
> Table 3 some lowercase first letters
>
> **Response to Clarity:**
> Thank you for your careful comments. We have fixed them in the revision.
>
> **Relation To Prior Work:**
> As stated above, I hope there be more related work in current status and future application of privacy policy study.
>
> **Response to ''Relation To Prior Work'':**
> Following your valuable suggestions, we incorporated more related work and future application of privacy policy study in Section 2 in the revision.
>
> **Documentation:**
> The construction process is well documented, but adopting law experts is itself a big human factor that might deter the replication of the study in other language or domain. I hope there be a guideline for reducing the variance coming from such human factors.
>
> **Response to Documentation:**
> Subjective factors from law  experts may interfere with the annotation process. To evaluate the subjective factors, we randomly select 20 privacy policies, which are jointly annotated by three legal experts, resulting in a Cohen Kappa coefficient of 0.907. This result strongly validates the annotation consistency across legal experts. We may develop annotation guidelines that define the criteria for annotations and provide concrete examples. For future work, we may develop annotation guidelines that define the criteria for annotations and provide concrete examples. We may also include non-experts in the annotation process with proper guidance considering perspectives that are closer to the end-users of the summarized policies. We added the corresponding discussion in Section 3.1 in the revision.

---

> ### Author Response · Authors · 2023-08-21
>
> **Ethics:**
> Only concern I have is the availability of the source data (document from the app stores of the private company) and hope authors checked the usability thoroughly.
>
> **Response to Ethics:**
> We made our Github repository including the source data public, you can access it by https://github.com/EnlightenedAI/CAPP-130.
>
> **Additional Feedback:**
> I wonder authors considered looking up privacy policies in Chinese law instead of ones in app stores, since I was first confused that the paper would handle privacy policy that applies like GDPR in European regions.
>
> **Response to Additional Feedback:**
> Our previous definition was not precise enough. A privacy policy serves as an online internet protocol crafted by service providers, which details how service providers collect, process, store, manage, and use personal information when users engage with applications. We modified the corresponding definition in the abstract and Section 1.
>
> Please let us know if these additions are satisfactory. Thank you for taking the time to review our paper and providing such valuable feedback.

---

### Official Review · Reviewer_wFwd · 2023-07-24
**Nice legal dataset in Chinese**

**Rating:** 6
**Confidence:** 4
**Correctness:** yes
**Clarity:** clear enough, some room for improvement.

**Strengths:**

Nice dataset; human expert annotations; useful summarizer.

**Additional Feedback:**

i would condition acceptance on seeing the code/replication materials.

**Documentation:**

the repo is empty.

**Limitations:**

Might be too niche for a general interest conference.

**Opportunities For Improvement:**

There could be more detail on how the corpus was collected.

The documents and summaries could also be provided translated in English.

Repo is empty.

**Relation To Prior Work:**

yes

**Summary And Contributions:**

This is a valuable dataset in Chinese legal domain, with 130 privacy policies, annotated by experts and also with paraphrases. They also train a summarizer.

---

> ### Author Response · Authors · 2023-08-21
>
> Thank you for your insightful and encouraging comments. We carefully and thoroughly revised the manuscript to address all the comments. The main changes are highlighted in blue and the detailed point-to-point responses are provided as follows.
>
> **Question 1:**
> There could be more detail on how the corpus was collected.
>
> **Response to Question 1:**
> We have introduced the source of the data, the strategy for collection, and the methodology for organization in Section 3. We further added the collection process in the section of ''Datasheets for Datasets'' in Appendix A.5.3.
>
> **Question 2:**
> The documents and summaries could also be provided translated in English.
>
> **Response to Question 2:**
> Thank you for your valuable suggestions. We are currently translating the documents and summaries into English. In addition, taking into account the differences in legal terms, privacy policy structures, and language habits in different countries and regions, we are customizing a tailored annotation scheme for English privacy policies and arranging for experts to annotate them.
>
> **Question 3:**
> Repo is empty.
>
> **Response to Question 3:**
> We made our Github repository public, you can access it by https://github.com/EnlightenedAI/CAPP-130.
>
> **Additional Feedback:**
> i would condition acceptance on seeing the code/replication materials.
>
> **Response to Additional Feedback:**
> We made our Github repository public, you can access it by https://github.com/EnlightenedAI/CAPP-130.
>
> Please let us know if these additions are satisfactory. Thank you for taking the time to review our paper and providing such valuable feedback.

---

### Official Review · Reviewer_qTDr · 2023-08-01
**CAPP-130 : A Corpus of Chinese Application Privacy Policy Summarization and Interpretations**

**Rating:** 6
**Confidence:** 5
**Correctness:** yes
**Clarity:** yes

**Strengths:**

Data Quality: The creation of the CAPP-130 corpus involved careful annotation and interpretation by legal experts. The use of multiple experts and the high Cohen's kappa coefficient (0.907) demonstrate the reliability and credibility of the annotations.

Performance Improvement: The experimental results show that TCSI-pp outperforms large language models like GPT-4 and other benchmark models in accurately extracting specific privacy terms and presenting them in a concise and understandable manner. This performance improvement is crucial in addressing the challenges of privacy policy comprehension.

Potential Impact: Privacy policies are crucial legal documents, and the proposed TCSI-pp framework has the potential to significantly improve their readability and user comprehension. By generating user-friendly summaries, it may empower users to make informed decisions about their privacy rights and avoid inadvertently agreeing to terms they do not fully understand.

Public Availability: The authors have made the CAPP-130 corpus, annotation guidelines, benchmark models, and source code publicly available on GitHub. This level of openness promotes reproducibility and encourages further research and development in privacy policy summarization.

**Additional Feedback:**

More evaluation and better visualization would improve the work.

**Documentation:**

yes

**Opportunities For Improvement:**

Readability Evaluation: While the submission claims improved readability and comprehension, the evaluation of readability is not extensive. It would be beneficial to conduct user studies or readability assessments to quantitatively measure the comprehensibility and effectiveness of the generated summaries.

Limited Negative Impact Assessment: Although the submission mentions that most companies hesitate to invest in summarization efforts for privacy policies, it does not delve into the potential negative societal impact of automated or semi-automated summarization of privacy policies. Ethical considerations, such as potential information loss or misinterpretation of clauses, should be thoroughly addressed to ensure that users' privacy rights are adequately protected.

Limited Comparison with Human Summaries: While the performance of TCSI-pp is compared to large language models and other benchmark models, a more comprehensive evaluation would involve comparing its outputs with summaries generated by human experts. A human baseline could provide a better understanding of TCSI-pp's effectiveness in achieving human-like summaries.

Annotation Bias: The creation of the CAPP-130 corpus involved annotations by legal experts, which may introduce bias based on the experts' interpretations and perspectives. While efforts have been made to ensure consistency, variations in interpretations may impact the quality of the corpus.

**Relation To Prior Work:**

yes

**Summary And Contributions:**

This work addresses the challenge of understanding complex privacy policies by proposing a Topic-Controlled Method for Privacy Policy Summarization and Interpretation framework (TCSI-pp) and introducing a fine-grained Chinese Application Privacy Policy Corpus (CAPP-130). Privacy policies are often written in verbose and technical language, leading users to agree to all clauses without understanding them, potentially compromising their privacy. Most companies are reluctant to invest in summarizing these policies. To tackle this, the authors introduce TCSI-pp, which generates user-friendly and comprehensible summaries. They fine-tune transformer models using CAPP-130 as a benchmark for data practice classification, summarization, and sentence rewriting. Experimental results show that TCSI-pp outperforms large language models like GPT-4 and other benchmarks in accurately extracting privacy terms and presenting them in concise language. However, TCSI-pp faces challenges such as intuitiveness and customization. Future work includes developing a user-friendly application and expanding TCSI-pp to support multiple languages and personalized summarization services for privacy policies.

---

> ### Author Response · Authors · 2023-08-21
>
> Thank you for your insightful and encouraging comments. We carefully and thoroughly revised the manuscript to address all the comments. The main changes are highlighted in blue and the detailed point-to-point responses are provided as follows.
>
> **Suggestion 1:**
> Readability Evaluation: While the submission claims improved readability and comprehension, the evaluation of readability is not extensive. It would be beneficial to conduct user studies or readability assessments to quantitatively measure the comprehensibility and effectiveness of the generated summaries.
>
> **Response to Suggestion 1:**
> Thank you for your valuable suggestions. To evaluate the readability of TCSI-pp-zh, we conducted a social survey on WJX(https://www.wjx.cn/). This survey consisted of 10 single-choice questions. Each question included a paragraph from a privacy policy along with four summaries generated by TCSI-pp-zh, CPT-4, LLAMA-2-Chinese, and Checkpoint (Randeng-Pegasus-523M-Summary-Chinese). We invited interviewees to select the summary they found most readable and comprehensible. From the survey, We eventually gathered 61 valid responses as shown in Table 1. Among the choices made by the interviewees, 46\% rated the readability of the summaries created by TCSI-pp-zh as the highest. TCSI-pp-zh received the most votes in six out of the ten questions.
> Moreover,  GPT-4 has the ability to extract specific details from privacy policies and generate summaries. However, it can also produce false content that is convincingly deceptive, containing information not found in the original text. In comparison, our approach is capable of summarizing and generalizing the extracted content, offering advantages such as accurate information extraction, strong interpretability, and concise understandability.
> We added the corresponding discussion in Section 4.4 and Section 4.5 in the revision and provided the statistics of the interviewees in Appendix A.4.
>
> Table1: Basic statistical information of the interviewees.
> | Model| Q1 |Q2 |Q3 |Q4|Q5 |Q6|Q7|Q8  |Q9|Q10|Percentage|
> |----|----|----|----|----|----|----|----|----|----|----|----|
> |TCSI-pp-zh|**37**|**29**|**29**| 17 |**29** |25|**38**| 24 |24|**30**|46.23%|
> |GPT-4|20|27|4|**39**|28|**31**|19|**30**|**35**| 24|42.13%|
> |Llama-2|	  4|4	|5   |3  |2  |5  |4   |5  |5  |5|6.39%|
> |Checkpoint|0	|1|23  |2  |2  |0  |0   |2  |2  |2|5.25%|
>
> **Suggestion 2:**
> Limited Negative Impact Assessment: Although the submission mentions that most companies hesitate to invest in summarization efforts for privacy policies, it does not delve into the potential negative societal impact of automated or semi-automated summarization of privacy policies. Ethical considerations, such as potential information loss or misinterpretation of clauses, should be thoroughly addressed to ensure that users' privacy rights are adequately protected.
>
> **Response to Suggestion 2:**
> Following your valuable suggestions, we tried to alleviate the problem of ''information loss and misunderstanding'' in the revision.
> Although we have made efforts in data annotation and model selection to deal with the possible ''illusion'' phenomenon of generative models, we still cannot completely eliminate the possibility of information loss and misunderstanding of terms. Therefore, on the basis of CAPP-130, we will explore the use of keyword extraction techniques to provide cues for classification models and establish external legal knowledge bases to provide valuable background knowledge for generative models. Furthermore, we may allow users to control the level of summarization or allow users to view both the summarized and original privacy policies side by side. In doing so, we hope to alleviate the problem of ''information loss and misunderstanding''. We added the corresponding discussion in Section 3.1 and Section 5.
>
> **Suggestion 3:**
> Limited Comparison with Human Summaries: While the performance of TCSI-pp is compared to large language models and other benchmark models, a more comprehensive evaluation would involve comparing its outputs with summaries generated by human experts. A human baseline could provide a better understanding of TCSI-pp's effectiveness in achieving human-like summaries.
>
> **Response to Suggestion 3:** Following your valuable suggestion, we incorporated human expert baseline in the revision. We found that the cross-paragraph summarization ability of TCSI-pp-zh is lower than that of legal experts. In the future, we will alleviate this issue by building a long-text summarization model. We added the corresponding discussion in Section 4.4 and Section 5.

---

> ### Author Response · Authors · 2023-08-21
>
> **Suggestion 4:**
> Annotation Bias: The creation of the CAPP-130 corpus involved annotations by legal experts, which may introduce bias based on the experts' interpretations and perspectives. While efforts have been made to ensure consistency, variations in interpretations may impact the quality of the corpus.
>
> **Response to Suggestion 4:**
> Following your valuable suggestion, we included a discussion on Annotation Bias in the revision. The creation of the CAPP-130 corpus involved annotations by legal experts, which may introduce bias based on their individual interpretations and perspectives. While efforts have been made to ensure consistency, such as strict annotation standards to improve Cohen's kappa coefficient of the annotations made by legal experts, variations in interpretations may impact the quality of the corpus. Furthermore, we may develop annotation guidelines that define the criteria for annotations and provide concrete examples. We may also include non-experts in the annotation process with proper guidance considering perspectives that are closer to the end-users of the summarized policies. We added the corresponding discussion in Appendix A.5.2.
>
> **Additional Feedback:**
> More evaluation and better visualization would improve the work.
>
> **Response to Additional Feedback:**
> Following your valuable suggestion, we incorporated a social survey in Section 4.5, as well as evaluation metrics like Micro-F1, Macro-F1, Bert-score, Bart-score, and Carburacy in the revision. We also highlighted the risk sentences to increase the visuality of the summarization.  We added the corresponding discussion in Section 3.2 and Section 4.
>
> Please let us know if these additions are satisfactory. Thank you for taking the time to review our paper and providing such valuable feedback.

---

### Decision · Program_Chairs · 2023-09-22

**Decision:**

Accept (Poster)

**Comment:**

This paper addresses the problem of Privacy Policy Summarization and Interpretation framework (TCSI-pp) and proposes a fine-grained Chinese Application Privacy Policy Corpus (CAPP-130). Careful manual annotations are included about different categories of privacy issues and reformulated sentences are provided. The dataset can be used for multiple experimental purposes: classification, summarization, and sentence rewriting.
The reviewers have found the dataset interesting and built with sufficient care. It provides a valuable resource for addressing an important questions of privacy policy, and complements well the existing resources.
The reviewers have raised several questions in their reviews. The authors have answered most of them in the rebuttal and made modifications in the paper accordingly.
Overall, this dataset is a valuable resource to the community for several tasks in the future.